# Objectively-Assessed Ultraviolet Radiation Exposure and Sunburn Occurrence

**DOI:** 10.3390/ijerph20075234

**Published:** 2023-03-23

**Authors:** Tammy K. Stump, Suzanne Fastner, Yeonjung Jo, Jonathan Chipman, Benjamin Haaland, Elizabeth S. Nagelhout, Ali P. Wankier, Riley Lensink, Angela Zhu, Bridget Parsons, Douglas Grossman, Yelena P. Wu

**Affiliations:** 1Department of Dermatology, University of Utah Health Sciences Center, Salt Lake City, UT 84132, USA; 2Huntsman Cancer Institute, University of Utah, Salt Lake City, UT 84112, USA; 3Department of Population Health Sciences, University of Utah, Salt Lake City, UT 84108, USA; 4Division of Public Health, Department of Family & Preventive Medicine, University of Utah, Salt Lake City, UT 84108, USA; 5Department of Pediatrics, University of Utah, Salt Lake City, UT 84113, USA

**Keywords:** skin cancer, ultraviolet radiation exposure, sunburn, prevention, melanoma

## Abstract

Ultraviolet radiation (UVR) exposure is the primary modifiable risk factor for melanoma. Wearable UVR sensors provide a means of quantifying UVR exposure objectively and with a lower burden than self-report measures used in most research. The purpose of this study was to evaluate the relationship between detected UVR exposure and reported sunburn occurrence. In this study, a UVR monitoring device was worn by 97 parent–child dyads during waking hours for 14 days to measure instantaneous and accumulated UVR exposure. The results showed that the participants’ total UVR exposure was associated with reported sunburn after adjusting for Fitzpatrick skin type and geographic location. It was observed that one standard erythemal dose (SED) increase in the participants’ daily total UVR exposure was associated with reported sunburn (an odds ratio (OR) of 1.26 with a 95% CI of 1.13 and 1.41, and *p* < 0.001 for parents and an OR of 1.28 with a 95% CI of 1.12 and 1.47, and *p* < 0.001 for children). A one-SED increase in the participants’ UVR exposure from 10 am to 4 pm was also associated with reported sunburn (an OR of 1.31 with a 95% CI of 1.15 and 1.49, and *p* < 0.001 for parents and an OR of 1.33 with a 95% CI of 1.12 and 1.59, and *p* = 0.001 for children). We found that elevated UVR exposure recordings measured by the UVR sensor were associated with reported sunburn occurrence. Future directions for wearable UVR sensors may include their use as an intervention tool to support in-the-moment sunburn prevention.

## 1. Introduction

In the United States, one in five adults will be diagnosed with skin cancer, making it the most commonly diagnosed malignancy [1]. The incidence rate of melanoma, the most deadly form of skin cancer, continues to increase, both within the United States and worldwide [2,3]. Although there are genetic and lifestyle factors that may predispose some individuals to skin cancer development, unprotected sun exposure is the main modifiable risk factor for skin cancers, including the deadliest form, melanoma [4,5,6]. Sun exposure, especially sunburns sustained during childhood, heightens the lifelong risk of cutaneous melanoma [7]. Even as rates of intentional indoor tanning decrease [8], sunburns remain common in both adults and children [9], with risk-taking attitudes likely contributing to sun exposure in some individuals [9,10], signaling a need for innovative approaches to promote sun protection behaviors (e.g., sunscreen use and seeking shade) that can reduce sun exposure.

Ultraviolet radiation (UVR) sensors can provide personalized information about an individual’s sun exposure and can be used as a standalone intervention or as an adjunct to more comprehensive interventions to reduce sun exposure and prevent sunburn. The tracking of objective UVR exposure is of high interest for melanoma-prone populations and has been shown to be an acceptable, nonreactive means of assessing sun exposure in observational and interventional studies [11,12]; however, there is limited research linking sensor-assessed UVR exposure and sunburns. An established association between sensor-assessed UVR exposure and reported sunburn would link sensor-assessed UVR to a clinically significant outcome, thereby providing additional support for the use of UVR sensors as an objective sun exposure measurement and intervention tool.

Sunburns and UVR exposure have been treated as independent outcomes in most past studies of sun exposure. The few studies that have simultaneously evaluated sensor-assessed UVR and the risk of sunburn have revealed important findings, but they also have some limitations. For instance, Xu et al. used wearable-sensor-detected UVR exposure to assess whether melanoma survivors exceeded their skin-type-specific minimal erythemal dose (i.e., the point at which burning is likely to begin) [13]. However, due to the low rates of sunburn in the sample, they could not directly compare sensor-assessed UVR and sunburn. In a larger study of over 340 Danish volunteers, Thieden et al. found that the sensor-assessed standard erythemal dose (SED) was higher on days when participants reported sunburns across a range of subgroups, including children, adolescents, indoor workers, and groups with outdoor hobbies (e.g., gardening) [14]. Taking place over a decade ago, this large study used paper diaries and UVR sensors that required manual downloads; these factors likely contributed to the missing and non-overlapping data noted. To improve upon the understanding of the relationship between sensor-assessed UVR and sunburns, there is a need for additional high-quality observational studies that employ current methods of assessing UVR, including electronic methods of data capture that allow for real-time data transmission. 

In addition to skin cancer prevention, wearable UVR monitoring has other potential applications which could benefit the public. Along with cutaneous melanoma, chronic UVR exposure is also associated with signs of photoaging and non-melanoma skin cancers, such as squamous cell carcinoma and basal cell carcinoma [15]. There is also some evidence that UVR exposure may contribute to uveal melanoma development [16]. Other than sunburn occurrence, acutely, individuals may experience photosensitivity reactions, particularly those with photodermatoses, such as porphyria cutanae tarda or lupus erythematosus, who are predisposed to photosensitive reactions, or photokeratitis [17,18]. Preventing these outcomes would be of interest to many members of the community, especially those who are at increased risk of skin cancer due to immunosuppression, which often arises as a result of solid organ transplantation, those taking photosensitive systemic medications, such as tetracyclines or naproxen, or those using photosensitive topical medications, such as acyclovir or hydrocortisone, that increase the risk of a photosensitive reaction [19]. Furthermore, individuals who are concerned about not receiving enough daily vitamin D may wish to use a UVR monitor to track their daily sun exposure.

The present study analyzes sunburn and UVR exposure data from a prior observational study in which parent–child dyads wore a Bluetooth-connected UVR sensor and completed online daily questionnaires to report sun protection use and sunburn development during outdoor activities over a 14-day period. This study included parent–child dyads due to the focus of the larger study on understanding relationships between parent and child sun behavior, and because these findings will be integral to building future sun protection interventions that would assist parents and children in co-managing sun exposure. Past analyses of this rich dataset have reported on reactivity to sensors [12], described the feasibility/acceptability of the procedures [20], evaluated the types of outdoor activities that dyads engaged in [21], and examined urban/rural differences in UVR exposure levels [22]. In the present study, we extend these findings by evaluating the extent to which sensor-assessed UVR predicts the risk of sunburn. Analyses of the relationship between daily total and peak UVR exposure and sunburn have been adjusted for skin type and rurality, two personal factors that have been found to be related to sun protection and exposure [22,23].

## 2. Materials and Methods

### 2.1. Participants

Participants were enrolled as dyads, consisting of one parent and one child. The total participation time of each dyad was two weeks, with data collection taking place from June to October. To be eligible, parents were required to be at least 18 years old; be the primary or sole caretaker for at least one child between 8 and 17 years old residing in the same household; live in Utah; and be willing to use a smartphone app to record UVR exposure data. Those who did not have access to a smartphone device with Wi-Fi and Bluetooth capabilities were provided a smartphone device by the study team to use for the duration of the study if they elected to participate. Participating children were required to be between the ages of 8 and 17 years old. Parents and children were ineligible for the study if they were taking sun-sensitizing medications or if they had a pacemaker (due to its possible interference with the UVR-monitoring device). Recruitment took place in urban and rural communities using multiple methods, including distribution of flyers at community and health promotion events, placing flyers in community settings (e.g., grocery stores), and mailing letters to potential participants through an online marketing resource. Of the 150 dyads screened, 116 were eligible and 97 parent–child dyads chose to participate.

### 2.2. Procedures

This study was based in Salt Lake City, Utah, United States (40.7608° N, 111.8910° W; altitude: 4226′), and included both urban and rural participants residing in the state of Utah. All study participation took place during the summer and early fall months for the area (June through October), when the UV index is high on most days [24]. All study activities were completed remotely, including the monitoring of UVR exposure over a 14-day period. Informed consent or assent was obtained from all parent and child participants, respectively. All procedures were approved by University of Utah’s Institutional Review Board. Both members of the participating dyad were asked to wear the Shade UVR monitoring device clipped to their clothing on their upper chest near the lapel, during waking hours for the entire 14-day duration of their participation [25]. Additionally, they were asked to complete surveys: (1) at baseline, (2) at the end of each day during the 14-day observation window, and (3) immediately following the 14-day observation window. The baseline and immediate post-study surveys included questions on participants’ sun protection behaviors (e.g., sunscreen use) and recent sunburn occurrences. The daily surveys asked details about participants’ outdoor activities that day and any sunburn occurrence. For all surveys, parents were asked to report on themselves and on their participating child. Children were asked to report answers for themselves only. Responses were recorded in REDCap [26]. The parent and child participants were compensated for their participation with gift cards up to a total of USD 50.

### 2.3. Measures

All self-report measures are described below and included in Appendix A.

Demographic characteristics. Parents were asked to report their demographic characteristics for themselves and their child at the start of the study, including age, sex, marital status, level of education, race, ethnicity, family income (less than USD 60,000 or greater than or equal to USD 60,000), and occupation environment (mainly indoors, mainly outdoors, in a motor vehicle, or other). Rural or urban home designation was categorized using the Rural Urban Commuting Area (RUCA) code [27].

Fitzpatrick skin type. Parent and child skin type was assessed by a set of questions asking about physical characteristics (i.e., eye color, hair color, and skin color, the propensity of the skin to tan or burn, and the sensitivity of the face to the sun) [28].

Daily sunburn. A daily survey question assessing sunburns was adapted from the Sun Habits Survey [29]. Each evening, participants were asked whether or not they experienced a red or painful sunburn that day.

UVR exposure. The Shade device was the UVR sensor worn daily during the participants’ two-week enrollment in the study. This device records UVR exposure in the units of J/m^2^, which can then be converted to the standard erythemal dose (SED) for analysis [30]. Specifically, the Shade UVR sensor (model V1.00, YouV Labs Inc., New York, NY, USA) is a wearable sensor that is part of a patented system in which Bluetooth Low Energy is used to connect the sensor to the Shade mobile app. The Shade UVR device maintains an internal data log of UVR dose (J/m^2^) accumulated every 6 min. In comparison to laboratory-grade devices, the Shade UVR sensor has demonstrated accuracy and sensitivity that are superior to other commercially available devices [25]. At the end of each day, participants were asked to sync their Shade device with a Shade smartphone application to confirm transmission. The application did not provide any feedback directly to participants about their daily UVR exposure or sun habits. Synced data could be accessed by the research team for analysis via a HIPAA-compliant cloud-based server.

Statistical analysis. We used logistic generalized estimating equation (GEE) models [31] to assess the association between daily UV exposures (total and peak) and sunburn. Sunburn is a binary outcome measured at the end of each day of the study, and daily UV exposure is a repeated measure over the 14 days of the study. UV exposure is clustered by individual patients, and an exchangeable correlation structure with robust standard errors was used to take the clustering by patients into account. The exchangeable correlation structure assumes that a patient’s UV exposure throughout the week is equally correlated, and the robust standard errors allows for departures from this assumption. From the GEE model, we estimated how the odds of sunburn increase per one SED increase in UV exposure that we adjusted for Fitzpatrick skin type and rural versus urban location. The analysis was done separately in children and parents. We also tested for the interaction effects of UV exposure and rural or urban location. The same set of analyses were performed with total UVR during peak hours as the exposure. All statistical analyses were completed using R software (version 4.2.1) with a statistical significance set to *p* ≤ 0.05.

## 3. Results

Of the 97 participating dyads, 56 (57.7%) were located in a rural location, 73 (75.3%) had a participating parent who was female, and 56 (57.7%) had a participating child who was female. The average parent age was 41.6 years (SD = 6.3), and the average child age was 12.7 years (SD = 2.7). In terms of race and ethnicity, 87% of adults and 85% of children were non-Hispanic White, 5% of adults and 8% of children were Hispanic, 5.3% of parents and 4.2% of children were Asian or Asian American, and 2.1% of adults and children self-identified as “other” for race. Daily surveys were completed by the vast majority of participants, with 84% of adults and 75% of children completing all their daily surveys. Two dyads withdrew from the study. On average, parents reported wearing the UVR sensor 12.7 (SD of 2.54) out of 14 days and children reported wearing the device 12.2 (SD of 3.03) out of 14 days. Regarding sensor wear, 73% of parents and 61% of children reported that they wore the device any time that they were outside from 7 am to 7 pm throughout the duration of the study.

### Objectively Measured UVR Exposure and Sunburn Occurrence

Daily total UVR exposure. The participants’ daily total UVR exposure was associated with reported sunburn in both parents and children after adjusting for Fitzpatrick skin type and geographic location (rural vs. urban). As shown in Figure 1, we observed that a one-SED increase in parents’ daily total UVR exposure was associated with increased odds of sunburn by a factor of 1.26 (95% CI: 1.13 and 1.41; *p* < 0.001) and that a one-SED increase in children’s daily total UVR exposure was associated with increased odds of sunburn by a factor of 1.28 (95% CI: 1.12 and 1.47; *p* < 0.001). There was suggestive evidence that the effect of UVR exposure was greater for rural children than urban children (*p =* 0.075). For rural children, the odds of sunburn increased by a factor of 1.48 (95% CI: 1.17 and 1.88; *p =* 0.001) per one-SED increase in daily total UVR exposure, compared to 1.12 (95% CI: 0.99 and 1.27; *p =* 0.06) for urban children.

Daily peak UVR exposure. The analysis of daily peak UVR exposure revealed comparable findings to those observed for total UVR exposure. Daily UVR exposure in the peak hours (10 am–4 pm) was also associated with reported sunburn occurrence among both parents and children after adjusting for Fitzpatrick skin type and geographic location (rural vs. urban). As shown in Figure 2, a one-SED increase in parents’ daily peak UVR exposure was associated with a 1.31 factor increase in the odds of sunburn occurrence (95% CI: 1.15 and 1.49; *p* < 0.001). Similarly, a one-SED increase in the children’s daily peak UVR exposure was associated with a 1.33-fold increase in the odds of a reported sunburn (95% CI: 1.12 and 1.59; *p =* 0.001).

## 4. Discussion

Wearable UVR sensors offer the potential to quantify sun exposure objectively and more accurately than self-report measures, which can be subject to recall and social desirability biases. The utility of these devices for measurement and intervention purposes depends on whether sensor-assessed UVR levels are associated with clinically meaningful outcomes, such as sunburn. In this study, UVR sensors and questionnaires were used to monitor UVR exposure among parent–child dyads. The simultaneous recording of UVR exposure from these two data sources enabled the evaluation of the relationship between daily total and peak sensor-assessed UVR exposure and reported sunburn occurrence. The results showed that sensor-assessed UVR exposure was associated with a statistically significant increase in sunburn risk. The magnitude of the effect on sunburns was essentially equivalent when measuring the total daily and peak-hour UVR and when adjusting or not adjusting for location and Fitzpatrick skin type. These findings support the utility of UVR sensors for measuring and approximating clinically meaningful outcomes in the context of skin cancer prevention, bolstering arguments for the potential utility of UVR sensors for observational and intervention purposes.

Importantly, the relationship between UVR exposure and sunburns remained statistically significant after controlling for skin type and rural (vs. urban) location—two factors that are known sources of variability in the study sample and that are linked to sunburn risk [22,23]. These findings suggest that UVR sensors may be beneficial for predicting sunburn, regardless of the individual’s skin type and location (i.e., urban vs. rural). The inclusion of these control variables also revealed suggestive evidence that the relationship between sensor-assessed UVR exposure and participant-reported sunburn was stronger for rural, compared to urban, residents. This finding is consistent with prior research indicating that rural residents tend to spend more time outdoors and use sun protection less frequently than their urban counterparts [32,33,34].

UVR sensors hold promise as tools that can provide researchers and wearers with real-time information that predicts sunburn risk. Through on-device display, near-field communication (NFC), and/or Bluetooth connection, UVR sensors can provide wearers with information on both their instantaneous and accumulated UVR exposure [35]. Recent interventions that have included sensors to provide feedback on UVR exposure have yielded reductions in unprotected sun exposure among adults and adolescents [36,37,38]. Given our finding that sensor-assessed UVR exposure predicted sunburn in children, UVR sensors may additionally be an effective skin cancer prevention tool for both children and their parents. For instance, UVR sensors could be used to notify the parent and child when the child’s accumulated UVR has reached unsafe levels or to help promote sun-safe decisions and planning.

Personally worn UVR sensors are an important addition to the landscape of UVR measurement tools, and more generally, to the assessment of behaviors related to skin cancer prevention. As summarized in recent reviews [39,40], the assessment of UVR using personally worn devices began over 50 years ago. In these early efforts, polysulphone badges were utilized and color changes signified UVR exposure. With even this relatively coarse measurement, researchers were able to evaluate several important questions, including the link between sun-exposed body sites [41] and skin cancer and the impact of varied occupations on sun exposure [42]. Personal UVR exposure measurement was also revealed to vary significantly based on the area of the body to which badges were attached [43]; this issue remains an important limitation for all worn UVR measurement tools, including the sensor used in this study. For this project, we reduced confounding factors across the participants by keeping the location of the worn sensor consistent (i.e., on the upper chest or near the lapel). Electronic dosimeters (e.g., a SunSaver wrist watch) began to be used for UVR measurement about 25 years ago and offered the advantage of providing time-stamped measurements of exposure as well as enabling exposure over longer periods of time [44,45].

Whereas personally worn sensors can provide personalized, real-time measurements of UVR exposure, it should also be noted that there are alternative UVR measurement approaches which can be considered depending on the purpose and scope of a particular study. Self-report questionnaires and geographical location have been used to infer UVR exposure, particularly when exposure over longer periods of time or at the population level is of interest [46,47]. Skin color can also be used to infer UVR exposure. For instance, skin reflectance measurements can be taken by a spectrophotometer in order to estimate the CIE (Commission Internationale de l’Eclairage) L*, a*, and b* colorimetric parameters representing the luminance, redness, and yellowness of the skin, respectively. These values can be used to evaluate the overall skin type of individuals, and the change in these values over time can be evaluated as a measure of change in UVR exposure [48,49]. For the present study, we determined that Bluetooth-connected UVR sensors were the best fit for the research questions. The goal of the overall study was to assess instances of sun exposure in-the-moment and to identify the specific activities and conditions that were associated with the greatest UVR exposure. In the analyses reported here, we examined how UVR exposure predicted daily sunburn occurrence; UVR sensors provided the personalized and fine-grained level of information needed to assess this relationship on the daily level.

The current study has several important limitations to acknowledge. Firstly, this study included a sample with relatively low racial/ethnic diversity, with over 85% of parents and children identifying as non-Hispanic White. Less than 10% of parents and children self-identified as Hispanic or Asian/Asian American and there was no Black or African American representation in parents or children. The demographics of the study sample reflect those of the geographic area, which has high rates of melanoma. Secondly, sunburns were assessed through a single item on daily surveys, which relied on self-report and included a limited assessment of sunburn severity. Further, our measurement and analytic strategy assumed that sunburn would be apparent on the same day as exposure, although erythema can take hours to become visible following exposure [50,51]. In future studies, questionnaires could include more detailed questions on the timing and degree of sunburn developed, and participants could be asked to include a photograph of reported sunburns. Thirdly, while adherence to wearing the sensor was good, there were still some missed measurements of UVR exposure due to challenges including difficulty remembering to wear the UVR sensor, keeping the device uncovered by clothing, or keeping the device securely attached to clothing [20]. Fourthly, the models reported here included a small subset of the variables that could impact the relationship between UVR and sunburns. In future studies, sun protection behaviors, risk behaviors, and/or weather-related factors (e.g., cloud cover) could be included to learn more about how these variables impact individuals’ sunburn risk.

## 5. Conclusions

Sensor-assessed UVR exposure among adults and children predicts sunburn. This association was held across several scenarios: for both parents and children, total and peak exposure, and with and without controlling for skin type and rural (vs. urban) location. The results showed a consistent association between sensor-assessed UVR exposure and the clinically significant outcome of sunburn, thereby providing additional support for the use of UVR sensors as an objective sun exposure measurement and intervention tool.

## Figures and Tables

**Figure 1 ijerph-20-05234-f001:**
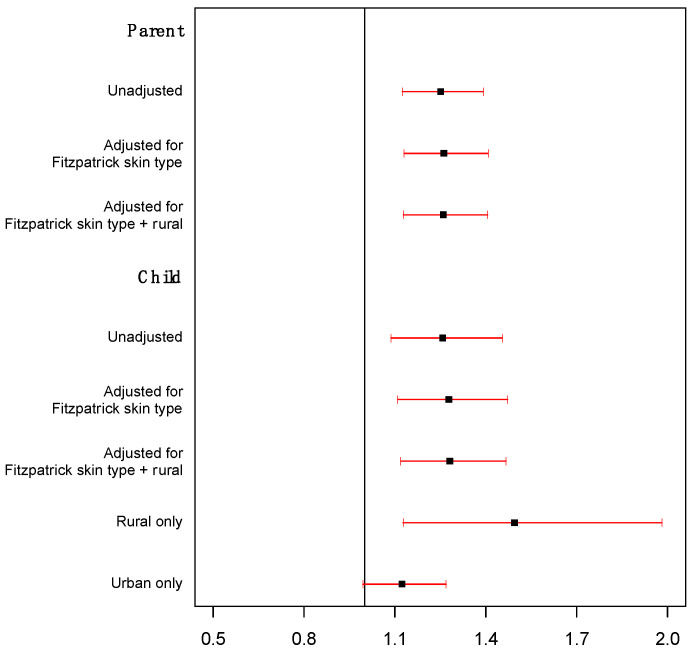
Hazard of sunburn in relation to daily total UV exposure.

**Figure 2 ijerph-20-05234-f002:**
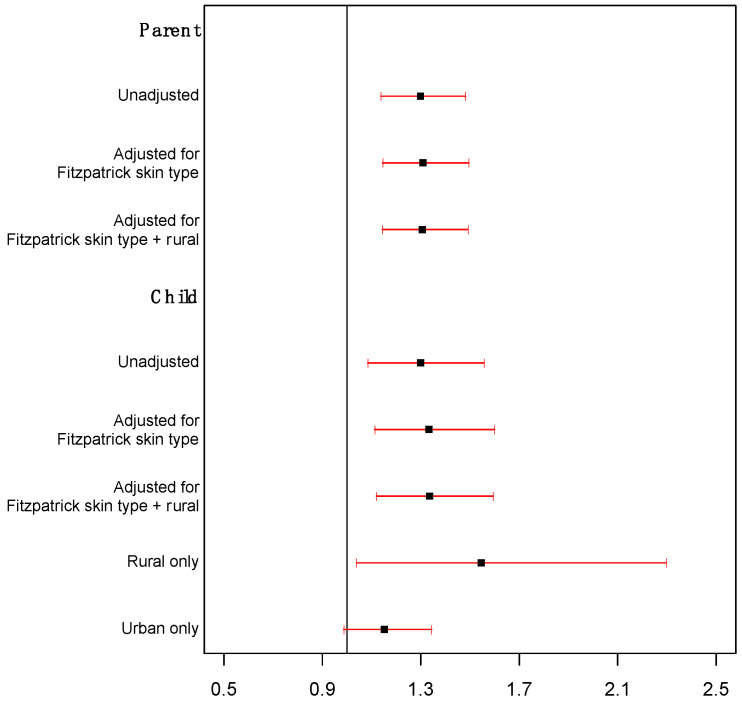
Hazard of sunburn in relation to daily peak UV exposure.

## Data Availability

The datasets generated during and/or analyzed during the current study are available from the corresponding author on reasonable request.

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
