# Peer review of "Objectively-Assessed Ultraviolet Radiation Exposure and Sunburn Occurrence"

_ijerph, 2023, doi:10.3390/ijerph20075234_

Round 1

Reviewer 1 Report

The authors present a study on the relationship between UVR detected exposure and sunburn occurrence. The focus is clearly on easy of use of weareble UV device as a auxiliar photoprotection tool, and for sunburn and skin cancer prevention.  As expected, they conclude that total UVR exposure was associated with reported sunburn. 

While the paper is mostly well-written and the introduction provides a relevant overview, there are also some problems. In it is current state, it seems the scientific value of the paper is mainly in showing that sunburn can be predicted/prevented with decent accuracy using UVR devices. However, details are lacking in what assumptions were used when generating the data were clustered. 

1) Please, provide more detailed information on the use of GEE for adjusting UVR exposure and sunburn (statistical analysis section). 

2) Please, provide (in an annex or in a web-repository) the survey form with the questions asked to the volunteers. 

3) It was observed that one standard erythemal dose (SED) increase in participants’ daily total UVR exposure was associated with reported sunburn. However, information on the characteristics of UVR in the study region is lacking  (e.g. Increase of 1 SED relative to which UVR levels?). Please provide more details on observed UVR levels (UV index), seasons of year, atmospheric conditions, geographic information of the measurement location (lat, lon, altitude), etc.

4) I didn't notice any significant differences between the adjusted and unadjusted data (Fig. 1). Please clarify.

5) Authors have concluded that wearable UVR sensors offer the potential to quantify sun exposure objectively and more accurately than self-report measures. I agree that self-report observations can be subject to recall and social desirability biases. However, some details were missing: Was there any evaluation of the quality of the answers: i) what defined a correct answer regarding sunburn?; ii) what defined an answer about the adequate use of sunscreen?; iii) what is the period of time to define the occurrence of sunburn?(lines 105/106)

I recommend a major review to address my concerns, after which the paper may be publishable. 

Author Response

1. Please, provide more detailed information on the use of GEE for adjusting UVR exposure and sunburn (statistical analysis section). 

We have now re-written our statistical analysis section to provide additional detail, including addressing your subsequent comments.

2. Please, provide (in an annex or in a web-repository) the survey form with the questions asked to the volunteers. 

We have now created an Appendix containing all study measures described in the methods section of this paper.

3. It was observed that one standard erythemal dose (SED) increase in participants’ daily total UVR exposure was associated with reported sunburn. However, information on the characteristics of UVR in the study region is lacking  (e.g. Increase of 1 SED relative to which UVR levels?). Please provide more details on observed UVR levels (UV index), seasons of year, atmospheric conditions, geographic information of the measurement location (lat, lon, altitude), etc.

We have now expanded upon our descriptions of the study’s geographical region and corresponding atmospheric conditions.

4. I didn't notice any significant differences between the adjusted and unadjusted data (Fig. 1). Please clarify.

We agree that the findings were quite similar between the adjusted and unadjusted GEE models. We now note this in the results and discussion.

5. Authors have concluded that wearable UVR sensors offer the potential to quantify sun exposure objectively and more accurately than self-report measures. I agree that self-report observations can be subject to recall and social desirability biases. However, some details were missing: Was there any evaluation of the quality of the answers: i) what defined a correct answer regarding sunburn?; ii) what defined an answer about the adequate use of sunscreen?; iii) what is the period of time to define the occurrence of sunburn?(lines 105/106)

Thank you for these inquiries. We have now enhanced the detail provided about study measures, including daily sunburn assessment. Verbatim measure text is now included in Appendix A.

Reviewer 2 Report

The authors present a contribution on objective evaluation of personal exposure to solar UV radiation in relation to sunburn occurrence.

The manuscript is in an important area, contributing to the scientific knowledge in human exposure. However, there are several issues that will need to be addressed before publication.

The introduction should be improved by enriching it with more studies. The methodology needs to be improved.

L36: The authors should  also give a general trend on the incidence of melanoma worldwide as reported in   Apalla, Z.; Lallas, A.; Sotiriou, E.; Lazaridou, E.; Ioannides, D. Epidemiological trends in skin cancer. Dermatol. Pract.

L37:  there are factors in addition to the cumulative UV exposure for developing skin cancers primarily the personal genotype (sensitivity, self-repair ability, etc.),  and age. An other factor is related to sun protection behaviour (see Morton et al, Risk attitudes and sun protection behaviour: Can behaviour be altered by using a melanoma genomic risk intervention? Cancer epidemiology, 61,8-13, 2019)

L38: the authors should also mention the detrimental effects related to long term solar UV exposure (premature skin aging, non-melanoma skin cancers, squamous and basal cell carcinoma) , as well as  acute effects consisting in erythema  photodermatoses, immunosuppression, phototoxicity/photoallergy and pigmentation (tanning) and in some eye pathologies. In addition, they should mention the beneficial role of vitamin D on human health.

L45: the following papers can be acknowledged to complete the information on solar UV exposure measurements:

Schmalwieser et al., Review on Occupational Personal Solar UV Exposure Measurements, Atmosphere 2021, 12, 142.

Schmalwieser, A.W.; Siani, A.M. Review on Nonoccupational Personal Solar UV Exposure Measurements. Photochem. Photobiol. 2018, 94, 900–915.

L54-L55: In some studies physical measurements of skin pigmentation were performed to evaluate the effect of UV exposure on skin color changes (for example Casale, G.R et al Extreme UV index and solar exposures at plateau Rosà (3500 ma.s.l.) in Valle d’Aosta Region, Italy. Sci. Total Environ. 2015, 512–513, 622–630).

L83-L95: More details are needed on how the authors recruited the participants. What is the rationale of considering dyads? The site location should be also given, as well as the duration on the use of devices. Who provided the smartphone device? Where was the device was worn? UV exposure is highly dependent on the anatomical site where the device is attached.

L99: More details on Shade UVR monitoring device are needed. Is these devices compared to UV reference instrument measuring ambient UV radiation? What is the spectral response of the device?  I suggest that the author adds references to similar devices, including a short review on the accuracy of sensors of this type.

L116: More details on daily survey sun-related behaviors should be provided.

L130: how did the authors evaluate the Fitzpatrick skin type of the participants ? How did they include this information to adjust daily exposure? Did the authors consider the tanning or if the participants have already tanned. How did the authors adjust the UV data for the type of location (rural or urban)?

More clarity is necessary in “An exchangeable correlation structure was used, which assumes observations within subjects are equally correlated”

L155: How did the authors evaluate that SED increase in parent’s daily total UVR exposure was associated with an increased odds of sunburn? The same for the following increased odds.

How did the authors take into account the posture of participants  and sky conditions in their analysis?

Author Response

  1. The introduction should be improved by enriching it with more studies. The methodology needs to be improved.

Thank you for this feedback. We have now added more studies to the introduction section in order to give more depth to the background to the research conducted.

  1. L36: The authors should also give a general trend on the incidence of melanoma worldwide as reported in   Apalla, Z.; Lallas, A.; Sotiriou, E.; Lazaridou, E.; Ioannides, D. Epidemiological trends in skin cancer. Dermatol. Pract. 

We appreciate this reference recommendation. We now describe the increasing incidence rate of melanoma both within the United States and worldwide within paragraph 1 of the introduction.

  1. L37:  there are factors in addition to the cumulative UV exposure for developing skin cancers primarily the personal genotype (sensitivity, self-repair ability, etc.),  and age. An other factor is related to sun protection behaviour (see Morton et al, Risk attitudes and sun protection behaviour: Can behaviour be altered by using a melanoma genomic risk intervention? Cancer epidemiology, 61,8-13, 2019)

We have now expanded upon the list of risk factors for skin cancer that is included in paragraph 1 of the introduction. We also now specifically reference sun protection behaviors as a method of reducing sun exposure.

  1. L38: the authors should also mention the detrimental effects related to long term solar UV exposure (premature skin aging, non-melanoma skin cancers, squamous and basal cell carcinoma) , as well as  acute effects consisting in erythema  photodermatoses, immunosuppression, phototoxicity/photoallergy and pigmentation (tanning) and in some eye pathologies. In addition, they should mention the beneficial role of vitamin D on human health.

Thank you for raising this important point. We agree that UVR monitoring is relevant for a range of conditions beyond skin cancer prevention and we have added a paragraph on this topic to the introduction.

  1. L45: the following papers can be acknowledged to complete the information on solar UV exposure measurements: 

Schmalwieser et al., Review on Occupational Personal Solar UV Exposure Measurements, Atmosphere 2021, 12, 142. 

Schmalwieser, A.W.; Siani, A.M. Review on Nonoccupational Personal Solar UV Exposure Measurements. Photochem. Photobiol. 2018, 94, 900–915.

Thank you for recommending these papers. We have now enhanced our overall discussion of UVR measurement, and have included citations from these reviews.

  1. L54-L55: In some studies physical measurements of skin pigmentation were performed to evaluate the effect of UV exposure on skin color changes (for example Casale, G.R et al Extreme UV index and solar exposures at plateau Rosà (3500 ma.s.l.) in Valle d’Aosta Region, Italy. Sci. Total Environ. 2015, 512–513, 622–630).

We agree that there are multiple methods of assessing UVR exposure. We now provide a discussion of other techniques of UVR monitoring and a justification for our focus on using UVR sensors in the current project.

  1. L83-L95: More details are needed on how the authors recruited the participants. What is the rationale of considering dyads? The site location should be also given, as well as the duration on the use of devices. Who provided the smartphone device? Where was the device was worn? UV exposure is highly dependent on the anatomical site where the device is attached. 

We have updated the procedures section to provide these additional details about the study. The discussion section also now includes information on variability in UVR measurement based on anatomical site.

  1. L99: More details on Shade UVR monitoring device are needed. Is these devices compared to UV reference instrument measuring ambient UV radiation? What is the spectral response of the device?  I suggest that the author adds references to similar devices, including a short review on the accuracy of sensors of this type. 

In the methods section, we now provide additional information about the Shade UVR monitoring device, including its accuracy and sensitivity.

  1. L116: More details on daily survey sun-related behaviors should be provided.

We now clarify that sunburns alone were analyzed in this study; other sun-related outcomes or behaviors from this study are reported in prior publications. All measures mentioned in the measures section are include in the newly created Appendix.

  1. L130: how did the authors evaluate the Fitzpatrick skin type of the participants ? How did they include this information to adjust daily exposure? Did the authors consider the tanning or if the participants have already tanned. How did the authors adjust the UV data for the type of location (rural or urban)?

We have now expanded on our description of the of the self-report measures used to assess Fitzpatrick Skin Type. We have also re-written our methods to emphasize that we adjusted our GEE model for location type to address this question.

  1. More clarity is necessary in “An exchangeable correlation structure was used, which assumes observations within subjects are equally correlated”

We have re-written the analysis section to focus on how the correlation structure is related to UV exposure throughout the study.

  1. L155: How did the authors evaluate that SED increase in parent’s daily total UVR exposure was associated with an increased odds of sunburn? The same for the following increased odds.

In our revision, we provide more clarity and details regarding the fact that UV is measured per SED increase. We also note that we test the coefficients at p <= 0.05.

  1. How did the authors take into account the posture of participants  and sky conditions in their analysis?

This is an interesting point. Our analyses did not account for posture of participants or sky conditions given that participants were asked to wear the UV monitors continuously over the course of time in their daily lives for 2 weeks. We agree that these can be important factors to consider in UVR exposure measurement. The discussion now includes mention of these variables.

Round 2

Reviewer 1 Report

The authors carefully considered my comments and suggestions, improving the manuscript. However, Appendix A was cited but needed to be shown. Please, add Appendix A to the paper. 

The authors can make changes to adequately address this lack in a minor revision, after which another round of review is probably unnecessary, and the paper should be ready for publication.

Author Response

Appendix A has now been directly added to the manuscript.

Reviewer 2 Report

The authors have adequately answered to the issues of the my review.  I would only recommend to report all the authors of the papers of the reference list

Author Response

We have now changed the reference style to MDPI, thereby showing all author names.